# Dietary Protein Intake Level Modulates Mucosal Healing and Mucosa-Adherent Microbiota in Mouse Model of Colitis

**DOI:** 10.3390/nu11030514

**Published:** 2019-02-28

**Authors:** Sandra Vidal-Lletjós, Mireille Andriamihaja, Anne Blais, Marta Grauso, Patricia Lepage, Anne-Marie Davila, Roselyne Viel, Claire Gaudichon, Marion Leclerc, François Blachier, Annaïg Lan

**Affiliations:** 1UMR PNCA, AgroParisTech, INRA, Université Paris-Saclay, 75005 Paris, France; sandravidalll@gmail.com (S.V.-L.); mireille.andriamihaja@agroparistech.fr (M.A.); anne.blais@agroparistech.fr (A.B.); marta.grausoculetto@agroparistech.fr (M.G.); anne-marie.davila-gay@agroparistech.fr (A.-M.D.); claire.gaudichon@agroparistech.fr (C.G.); francois.blachier@agroparistech.fr (F.B.); 2UMR MICALIS, AgroParisTech, INRA, Université Paris-Saclay, 78350 Jouy-en-Josas, France; patricia.lepage@inra.fr (P.L.); marion.leclerc@inra.fr (M.L.); 3H2P2, Biosit-Biogenouest, Université de Rennes 1, 35005 Rennes, France; roselyne.viel@univ-rennes1.fr

**Keywords:** dietary protein level, colitis, epithelial repair, mucosa-adherent microbiota

## Abstract

Mucosal healing after an inflammatory flare is associated with lasting clinical remission. The aim of the present work was to evaluate the impact of the amount of dietary protein on epithelial repair after an acute inflammatory episode. C57BL/6 DSS-treated mice received isocaloric diets with different levels of dietary protein: 14% (P14), 30% (P30) and 53% (P53) for 3 (*day 10*), 6 (*day 13*) and 21 (*day 28*) days after the time of colitis maximal intensity. While the P53 diet worsened the DSS- induced inflammation both in intensity and duration, the P30 diet, when compared to the P14 diet, showed a beneficial effect during the epithelial repair process by accelerating inflammation resolution, reducing colonic permeability and increasing epithelial repair together with epithelial hyperproliferation. Dietary protein intake also impacted mucosa-adherent microbiota composition after inflammation since P30 fed mice showed increased colonization of butyrate-producing genera throughout the resolution phase. This study revealed that in our colitis model, the amount of protein in the diet modulated mucosal healing, with beneficial effects of a moderately high-protein diet, while very high-protein diet displayed deleterious effects on this process.

## 1. Introduction

In recent years, mucosal healing (MH) has become a therapeutic goal for the prevention of the complications of inflammatory bowel diseases (IBD) [1,2,3,4]. MH relies on concomitant cellular events that provide epithelial repair to restore the tissue integrity and the physiological functions of the colon such as barrier function and water absorption [4,5], these events being presumably protein and energy consuming processes. In addition, the nutritional status is known to be impaired during and after an intestinal inflammatory episode due in particular to the catabolic action of the pro-inflammatory cytokines [6]. Although there is some evidence that the daily protein needs of IBD patients may differ from those of healthy controls, further experimental and clinical data are required to fully validate this concept. The European Society for Clinical Nutrition and Metabolism recommends to increase the protein intake in active IBD to 1.2–1.5 g/kg body weight/day in adults and to maintain a similar level as it is recommended for the general population during remission (about 1 g/kg body weight/day) [7]. However, the beneficial effects of increasing dietary protein level have been barely studied. Notwithstanding, majoring dietary protein intake may impact the colonic mucosa in the healthy gut as recently reviewed [8] by altering colonocyte biology [9,10] and colonic luminal environment [11,12,13,14]. Furthermore, we showed that a high protein (HP, 53% of total energy provided by protein) intake exacerbated acute colonic inflammation when simultaneously given with colitis induction [15]. Interestingly, the same study showed that this protein level in the diet was rather beneficial after the acute episode of colitis as it enhanced colonic epithelium restoration in surviving animals, thus indicating a potential effect of the dietary protein level on colonic crypt repair after acute inflammation [15]. Actually, supplementation with some AA, alone or in combination, improved parameters related to mucosal healing in colitis model such as epithelial regeneration and colonic barrier function (for review, see Reference [16]). Moreover, published data suggest that the availability of some amino acids, such as threonine or cysteine, is limiting under conditions of colonic inflammation [17,18] and that dietary supplementation of protein or amino acids is needed to promote favorably epithelial repair. However, AA supplementation may also exert deleterious impact as it has been recently shown with threonine during colitis [19]. To date, the positive effect of any AA supplementation on colitis has never been demonstrated in clinical studies.

We then hypothesized that increasing dietary protein intake might influence colon epithelial repair by a direct effect of additional AA blood supply or via microbiota-derived effects. The present study thus aims to evaluate the impact of different level of dietary proteins (normoproteic (14% of the total energy provided by protein, P14)), moderately HP diet (30%, P30) and an elevated HP diet (53%, P53) on some relevant barrier function parameters after the acute inflammatory episode in the DSS-mouse model of colitis.

## 2. Materials and Methods 

### 2.1. Animals and Diets

Seven-week-old C57BL/6 male mice were obtained from ENVIGO, France (144, weight 18–23 g). Animals were housed in a 12:12-h light-dark circle at 23 °C and controlled humidity (55 ± 10%) and acclimated for one week with free access to standard mouse chow and tap water. Each mouse was maintained in an individual cage with a grid and was allowed to acclimate to the P14 diet after three days of a standard mouse chow/fresh P14 diet (Table 1). The study was performed according to the European directive for the use and care of laboratory animals (2010/63/UE) and received the agreement of the local animal ethics committee and of the ministerial committee for animal experimentation (registration number: APAFIS#3987-2016012214388658). 

### 2.2. Experimental Design

Three types of experimental isocaloric diets (14.5 kJ/g) were used in this study: a normoproteic diet (P14, 140 g/kg milk protein), a moderately HP diet (P30, 300 g/kg of milk protein) and an elevated HP diet (P53, 530 g/kg of milk protein) isocellulose diet with similar energy content (Table 1). Healthy controls (non-treated mice at *day 0*, *n* = 12) received fresh tap water. DSS-treated mice (*n* = 132) were given 3.5% (wt/vol) DSS (36,000–50,000 MW, MP Biomedicals Illkirch-Graffenstaden, France) in the drinking water for 5 days, from *day 1* to *day 5* (fresh DSS solution being prepared every two days) to induce an acute colitis episode (Figure 1). At *day 7*, corresponding to the maximal intensity of colon inflammation, mice were divided into three groups with the same mean body weight (BW) and were fed either the P14, the P30 or the P53 diet during 3, 6 or 21 days. Food and drink were given ad libitum and measured daily. Disease activity was scored based on stool consistency, rectal bleeding and percentage of body weight loss, as previously described [15]. For each parameter, a score of 0 to 3 was attributed: loss of weight (0, none; 1, 0%–10%; 2, 10%–20%; 3, >20%), stool consistency (0, normal pellets; 1, slightly loose feces; 2, loose feces; 3, watery diarrhea) and visible fecal blood (0, negative; 1, slightly bloody; 2, bloody; 3, blood in the whole colon). Mice were euthanized to evaluate the impact of diet on epithelial repair kinetics at *days 10* (*n* = 12), *13* (*n* = 12) and *28* (*n* = 16) per diet group (Figure 1). Body fat and lean body mass of mice belonging to *day 28* group, were measured every nine days with dual-energy x-ray absorptiometry (DEXA) using a PIXImus imager (GE Lunar PIXImus, GE Healthcare, Fitchburg, WI, USA). Mice were euthanized if they lost more than 20% BW, as per approved animal protocol guidelines, to meet the end point criteria. 

### 2.3. Intestinal Permeability Assessment

Firstly, epithelial barrier integrity was assessed in vivo with the 4 kDa paracellular marker fluorescein isothiocyanate (FITC)-labeled dextran (FD4, Sigma-Aldrich, St Quentin Fallavier, France) at *days 9*, *12* and *18*. Food was withdrawn for 4h before gavage (600 mg/kg BW of FD4) and plasma was collected from the tail 4 h later (*n* = 8/diet group). Fluorescence intensity of each sample was later used to calculate permeability using standard curves generated by serial dilution of FD4. Secondly, at euthanasia, the proximal colon section was mounted in EasyMount Ussing chambers (Physiologic Instrument Inc, San Diego, CA, USA), within 15 min from dissection as previously described [13]. Paracellular permeability was assessed by measuring the mucosal-to-serosal flux of FD4 at a final concentration of 0.25 mg/mL, and fluorescence units (FU) were measured 90 min later with the Infinite^®^ 200 Pro spectrofluorimeter (TECAN, Männedorf, Switzerland). Tissue viability was assessed at the end of each recording by adding the cholinergic drug carbachol (10^−4^ M) on the serosal side. 

### 2.4. Tissue Collection

Mice were euthanized by an intracardiac puncture after sedation by isoflurane. Plasma was frozen and kept at −80 °C for later measurement of cytokines and lipopolysaccharide binding protein (LBP) concentrations. Colon was resected, measured, weighted and the proximal colon section mounted in Ussing chamber. Proximal colon mucosa was scraped for latter mucosa-adherent bacterial DNA extraction procedure. Colon samples were harvested for RNA analysis, myeloperoxidase (MPO) activity and protein expression assays, immediately frozen in liquid nitrogen and stored at −80 °C after resection. Histological analysis was performed with distal colon fixed in 4% buffered formaldehyde. 

#### 2.4.1. Determination of Local and Systemic Inflammatory Markers

Colon inflammation was assayed with MPO assay measurement as described in Reference [20] and colonic IL-1β and IL-6 concentrations were measured by Luminex technology in total colon protein lysate by using Bio-Plex kits (Bio-Rad, Marnes-La- Coquette, France). Plasma concentration of LBP was determined with a commercial solid-phase sandwich ELISA (PikoKine ELISA Kit Mouse, Set EK1274; Boechout, Belgium).

#### 2.4.2. Histological Analysis

Histological and re-epithelization scores on hematoxylin-and-eosin (HE) stained colonic sections were calculated as previously described (Vidal-Lletjós, et al., submitted) after blind microscopic assessment performed by the histological platform Histalim (Montpellier, France). Quantitative evaluation of well-oriented crypts and cell numeration in periodic acid-schiff (PAS) stained 4-µm transversal colon sections was determined using the image analysis software Pannoramic Viewer v. 1.15.4 (3DHISTECH, Budapest, Hungary). Paraffin-embedded distal colon samples were cut into 4 µm-thick sections, mounted on positively charged slides and dried. Immunohistochemical stainings of Ki67 and Caspase 3 were performed on the Discovery XT Automated IHC stainer using either Ventana DAB MAP detection kit (Ventana Medical Systems, Tucson, AZ, USA) or the Ventana CHROMO MAP detection kit (Ventana Medical Systems, Tucson, AZ, USA). Following deparaffination with Discovery wash solution (Ventana), antigen retrieval was performed using a Tris-based buffer solution. Later, endogen peroxidase was blocked and slides rinsed before incubation with primary antibodies either rabbit anti-ki67 (NB600-1252 Novusbio, Centennial, CO, USA) diluted at 1/100 or rabbit anti-Casp3 (9661, Cell Signaling, Danvers, MA, USA) diluted at 1/250. For Ki67 staining, signal enhancement was performed using Goat anti-Rabbit biotinylated secondary antibody (Vector laboratory, Burlingame, CA, USA) and DAB MAP detection kit. An anti-rabbit HRP (Ventana Medical Systems, Tucson, AZ, USA) secondary antibody and the CHROMO MAP detection kit (Ventana Medical Systems, Tucson, AZ, USA) was used for the caspase 3 staining. Slides were then counterstained with hematoxylin and rinsed. Slides were manually dehydrated and coverslipped. Ki67 labelling index was calculated as the percentage of Ki67 positive cells relative to the total number of cells within the same crypts.

#### 2.4.3. RNA Isolation and Quantitative Real-Time PCR

Total RNA was extracted after tissue homogenization in TRIzol^®^ Reagent (Invitrogen, Cergy Pontoise, France). Purification was performed with the RNeasy Mini Kit (Qiagen, Courtaboeuf, France) and a DNase step (Qiagen, Courtaboeuf, France). After cDNA synthesis from mRNA using a High Capacity cDNA Reverse Transcription Kit (Applied Biosystems, Fisher Scientific, Illkirch, France), quantitative real-time polymerase chain reaction (qRT-PCR) was performed (primer sequences available on demand) with the Fast SYBR Green MasterMix (Applied Biosystems, Fisher Scientific, Illkirch, France) and StepOne Real-Time PCR system (Applied Biosystems, Fisher Scientific, Illkirch, France). Gene expression level was normalized relative to the normalizing gene HPRT and normalized to *day 7* group with 2^−ΔΔCt^ calculation. 

#### 2.4.4. Western-Blot Analysis

Frozen colonic tissue was homogenized in a lysis buffer [15], and 25 µg of total protein lysates were loaded onto 4%–12% Criterion XT gel (Bio-Rad, Marnes-La-Coquette, France) before electrophoresis in MOPS buffer (Bio-Rad, Marnes-La-Coquette, France). After transfer onto nitrocellulose membrane and incubation in blocking solution (TBS pH 7.5, 0.05% Tween 20, and 5% (wt/vol) non-fat dry milk), membranes were incubated overnight (4 °C) with rabbit ZO-1 antibody (1/250, 617300, Invitrogen, Cergy Pontoise, France) or with mouse claudin-1 antibody (1/250, 374900, Thermo-Fisher Scientific, Bedford, MA, USA) diluted in blocking solution. After three washes, blots were incubated for 2 h at room temperature with an anti-rabbit or anti-mouse HRP-linked secondary antibody. Revealing was performed using enhanced chemiluminescence (ECL system, Pierce Biotechnology, Courtabœuf, France), and bands were quantified by densitometry using the FluorChem FC2 device and the AlphaView software (Cell Biosciences, Santa Clara, CA, USA). GAPDH expression (Ab9484, mouse monoclonal antibody, Abcam, Cambridge, UK) was used to ensure consistent protein loading and transferring. 

#### 2.4.5. Evaluation of Adherent Mucosal Microbiota Composition

From the scrapped adherent-mucosa, bacterial DNA extracts were obtained by using the PowerFecal DNA Isolation kit (MoBio Laboratories, Carlsbad, CA, USA) according to the manufacturer’s protocol. Specific regions of the bacterial 16S rDNA gene were amplified using real-time qPCR. Eubacteria were quantified by real-time qPCR using specific primers (HAD-1: 5′-TGGCTCAGGACGAACGCTGGCGGC-3′ and HAD-2: 5′-CCTACTGCTGCCTCCCGTAGGAGT-3′), annealing at 59 °C for total bacteria. In addition, the V3-V4 region amplification of the 16S rDNA gene was performed and sent to the GenoToul platform (Castanet-Tolosan, France) for MiSeq Illumina sequencing using the MiSeq kit V2 2 × 250 bp. Data quality check and analysis were performed using the INRA-Migale server, as previously published [21]. In parallel, sulphate-reducing bacteria group were quantified by real-time qPCR using specific primers targeting Dissimilatory Sulfite Reductase A (DSR-1F: 5′-ACSCACTGGAAGCACGGCGG-3′ and DSR-R: 5′-GTGGMRCCGTGCAKRTTGG-3′), Fast SYBR Green MasterMix (Applied Biosystems, Fisher Scientific, Illkirch, France) and StepOne Real-Time PCR system (Applied Biosystems, Fisher Scientific, Illkirch, France).

### 2.5. Statistical Analysis

All statistical analysis were performed with R software version 1.0.143 (Boston, MA, USA). For follow-up data, a mixed-model was used with mice as random effect while diet and time repeated factor were used as fixed effects. From *day 7*, the mean of values for each diet were compared using Bonferonni post-hoc tests. For parameters measured at euthanasia, Anova with diet as a fixed effect was used and the means were compared using Bonferonni post-hoc tests. For all statistical tests, the level of significance was set to *p* < 0.05. Principal component analysis (PCA) was performed on family-level taxonomy to assess the influence of diet on the mucosa-adherent microbiota composition at *days 10*, *13* and *28* as well as all times combined. The analysis of similarity (Anosim) test was used to assess the correlation between the ecological distance (based on family composition) and diet groups where an R-value of 0 indicates the highest dissimilarity possible and 1 indicates the highest similarity possible. 

## 3. Results

### 3.1. Dietary Protein Level Influences Colon Inflammation and Body Weight Recovery 

High protein diets, which were introduced the day of the maximal inflammatory score (at *day 7*), did not influence mortality, which was similar between the three groups (12.5% per diet) from *day 3* to *day 14*. After an average of 14% BW loss, mild to severe diarrhoea and rectal bleeding, disease activity index (DAI) started to decline after *day 8* for P14 and P30 diets (Figure 2A), while a two-day delay was observed for P53 mice. In this latter group of animals, the DAI was systematically two-point higher from *day 9* to *day 19* when compared to the other two diets (Figure 2A) due mainly to higher stool consistency score. This was confirmed by the area under the curve (AUC) values, even though P14 and P30 were not different between them (P14: 72.6 and P30: 71.03 AUC of DAI measured between 0 and 28 days, NS), both were statistically different from P53 (45.8 AUC of DAI for 28 days, vs P14 and P30, *p* < 0.01). In addition, P30 mice recovered their fat mass more rapidly than with the two other diets (P14: 33.9, P30: 49.2 and P53: 23.0 AUC of fat percentage evolution between *days 7* and *26*, *p* < 0.001) (Figure 2B). These differences were not related to the global energy intake level, since diet intake per day was similar whatever the diet (data not shown). P14 and P30 fed-mice presented a similar level of inflammation based on colon MPO activity (Figure 2C) and IL-6 and IL-1β concentrations (Figure 2D) but these inflammatory biomarkers were higher in P53-fed mice. Colonic MPO activity was indeed significantly increased at *day 28* for P53 mice compared to P30 (Figure 2C), as well as the pro-inflammatory cytokines at *day 10* and at *day 28* for IL-1β (Figure 2D). However, the overall histological score (based on crypt damage, ulceration and erosion, goblet cell depletion and cell infiltration) which showed high inter-individual variability, was not different between groups whatever the time (Figure 3A). Although the histological score was significantly reduced for all the groups when compared to *day 7* (9.3 ± 1.4, *p* < 0.05), some histological abnormalities persisted for all dietary groups at *day 28*. 

### 3.2. P30 Diet Enhances Epithelial Repair Features

P30 mice showed a significant thickening of the colonic wall at *day 10* (colon weight/length ratio: 0.037 ± 0.002) when compared to *day 7* (P30: 0.025 ± 0.002, *p* = 0.0059) and to P14 diet (P14: 0.030 ± 0.002, *p* = 0.0142). P30-fed mice were indeed the only group which recovered initial colonic length at *day 28* (*day 0*: 6.50 ± 0.17 cm vs *day 28* P30: 5.98 ± 0.16 cm, NS) contrastingly to other dietary protein groups (5.70 ± 0.20 cm). This increase in colon weight and length corresponded to a higher crypt height and a higher number of nuclei per crypt in P30 mice than in the P14 group at *day 10* (Figure 3B,C). The percentage of Ki67 positive cells was greater in the colonic tissue of P30 mice (vs P14, *p* = 0.0467), as illustrated in Figure 3D although 30% of the mucosa surface on average was devoid of crypts in all groups at *day 10*, and the number of caspase 3 positive cells was significantly reduced in P30 mice when compared to the other two diets (P30: 0.21 ± 0.07 vs P14: 0.70 ± 0.26 and P53: 0.98 ± 0.26 positive cells per crypt, *p* < 0.05). These epithelial morphometric changes coincided with increased gene expression of the epithelial repair factor *Tgf-ß1* in P30-fed mice (Table 2). Interestingly, the gene encoding Gpx2, which protects intestinal tissue against oxidative damage, displayed an increased expression in P30 mice at *day 10* (P30: 1.21 ± 0.21 vs P14: 0.78 ± 0.06 and P53: 0.51 ± 0.12, *p* < 0.05). In addition, the genes encoding *Tff3*, colonic *Saa* (Serum Amyloid A) and *Il-15* were also higher compared to P14 at *day 13* (Table 2).

### 3.3. P30 Diet Improves Dss-Induced Barrier Function Alterations

Interestingly, FD4 challenge showed an intestinal permeability reduction for animals receiving a P30 diet at *days 9* and *12* compared to the P14 group (Figure 4A) and this was confirmed at the colon level where FD4 flux measured in a Ussing chamber was decreased in P30 mice (7119 ± 507 FU) when compared to P14 (9945 ± 303 FU) at *day 10* (*p* = 0.014). This reduction of DSS-induced colon permeability was associated with lower LBP plasmatic concentrations at *days 13* and *28* compared to P14 and P53 groups (Figure 4B). In addition, the gene expression of the tight-junction proteins *Cldn1* and *Ocln* was differentially modulated by dietary protein intake. Indeed, *Cldn1* gene expression was increased by P30 at *days 10* and *13* while *Ocln* increased at *days 13* and *28* (Table 2). This result of *Cldn1* expression has been confirmed by western blotting (Figure 4A), where Claudin-1 protein level was higher in P30 colon at *day 13*. Moreover, ZO-1 was more expressed at *day 13* in P30 mice, when compared to the other two groups, although no significant difference at the gene expression level was detected (*Tjp1*, Table 2) and inter-individual variability was seen (Figure 4C). In addition, the level of dietary protein intake modulated colonic gene expression of several mucins. Indeed, both HP diets increased mRNA expression of the transmembrane mucin *Muc3* at day 13 and the main gel-forming mucin *Muc2* at *day 28* (which was also increased at *day 13* by the P53 diet only). In contrast, neither *Muc1* nor *Muc4* were affected by the diet (Table 2). 

### 3.4. DSS-Induced Dysbiosis is not Alleviated by Diet, but P30 Diet Increases the Proportions of Butyrate-Producing Bacteria

DSS-induced dysbiosis was not resolved within 28 days, whatever the diet (Table 3). Diversity and richness were still lower than at *day 0* (6.46 ± 0.07 and 794 ± 55.2 for Shannon and Chao indexes, respectively) and similar to *day 7* (4.75 ± 0.32 and 633 ± 43.7, for Shannon and Chao, respectively). Only the Simpson index was significantly less decreased in P30 group at *day 10* when compared to the two other diets. In addition, the number of observed OTUs at *day 13* was significantly decreased in P53 mice compared to P30 (Table 3).

The overall microbiota composition at the phylum level for each dietary group is shown in Figure 5A and the data describing microbial composition similarities based on family were compared between the diet groups with PCA with the three diet groups as classifying variables (Figure 5B–D). At *day 10*, mucosa-adherent microbial samples clustered by diet groups, as evidenced by their significant ecological distance (R = 0.32, *p* < 0.001) (Figure 5B). P53 group was mostly explained by *Clostridiales* (Firmicutes), *Enterobacteriaceae* (Proteobacteria), *Rikenellaceae* and *Porphyromonaceae* (Bacteroidetes) versus *Eubacteriaceae* (Firmicutes) and *Bifidobacteriaceae* (Actinobacteria) for P30 group (Figure 5B). *Haemophilus* (Proteobacteria) and *Alloprevotella* (Bacteroidetes) proportions were statistically increased in P53 mice at *day 10* while *Desulfovibrio* (Proteobacteria), an H_2_S producer, showed a higher abundance at *day 10* for P30 and P14 diets (vs P53, *p* < 0.05) (Table 3). However, the quantification of total sulphato-reducing bacteria did not indicate differences at any time for any diet (data not shown). Interestingly, the Bacteroides genus relative percentage was reduced in the P30 group (vs P14 and P53, *p* < 0.001) while caecal SCFA concentrations were higher (Table 3).

Although Firmicutes phylum proportions remained stable and similar to *day 0* (Figure 5) whatever the time and the diet, *Clostridium_XIVa* and *Faecalibacterium* proportions were higher in P30 group than in P53 at *day 13* (*p* = 0.005) (Table 3). In contrast to *day 10*, Bacteroidetes relative abundance was higher with both P30 and P53 diets owing to increased proportions of *Bacteroides* genus. Moreover, *Haemophilus* had increased proportions in the P53 group when compared to the P14 group at *day 13*, as well as *Enterorhabdus* (Actinobacteria) (Table 3). 

At *day 28*, the impact of dietary protein intake on mucosal-adherent composition was less marked between groups (Figure 5D) especially between P14 and P30 groups (Table 3). However, Actinobacteria, which proportions stayed low before *day 28* (*day 0–day 13*: 0.1%) (Figure 5A), were two-fold higher in P30 (P30: 3.1% vs P14: 1.4%, *p* = 0.014) and in the P53 group, the proportion of *Bifidobacteriaceae* being increased in that latter group (*p* = 0.021, P14: 0.60 ± 0.28%, P30: 1.70 ± 0.83% and P53: 0.93 ± 0.36%). In addition, the protein intake modulated microbiota activities since total caecal SCFA concentrations increased in both HP groups at *day 28*. Although Deferribacteres proportion was reduced for the three diets when compared to *days 10* and *13*, reaching *day 0* level (3.7 ± 1.0%, NS vs *day 28* whatever the diet), no other major differences at the phylum level were recorded. Interestingly, *Lachnospiraceae* family (Firmicutes) relative proportion at *day 28* was greater than the one at *day 7* (*day 7*: 13.48 ± 2.91% vs *day 28*: 24.22 ± 1.82%, *p* = 0.0184). Moreover, higher proportions of *Faecalibacterium* (*Clostridiaceae*) (vs P53) and *Roseburia* (vs P53 and P14) in P30 group were observed at *day 28* (Table 3). Contrastingly, P53 microbiota composition showed again noticeable taxonomic differences compared to P14 and P30 mice, with increased abundance of the genera *Porphyromonas*, *Ethanoligenens* and *Haemophilus* contrasting with a decreased abundance of *Bacteroides* (Table 3).

## 4. Discussion

The present study showed that the level of dietary protein intake differentially modulated mucosal healing after an acute colon inflammatory episode. While the P53 diet worsened both the intensity and duration of the inflammation induced by DSS treatment, the P30 diet, when compared to P14 diet, accelerated epithelial repair by favoring the restoration of colon barrier architecture and function. 

P30 diet indeed induced crypt hyperproliferation associated with increased gene expression of the repairing factors *Tgf-ß1* and *Tff3*, both factors being known to contribute to the integrity of mucosal surface continuity but in an independent manner [22]. In addition, *Saa*, which was over-expressed in P30 mice compared to the two other diets, was recently described as a protective factor against colon epithelium acute injury [23]. Altogether, the increased expression of genes encoding these promoting repair factors as well as tight-junction proteins (Claudin-1 and Zona-Occludens 1), likely contributed to the restoration of colon barrier integrity, as evidenced by lower permeability and bacterial translocation-related marker (LBP) in the systemic flow. Additionally, an increased mRNA level of *Gpx2*, which upregulation is associated with inflammation resolution [24], suggests a lower inflammation-induced oxidative stress in P30 animals. Furthermore, the P30-mediated positive effects on epithelial repair might be related to a modulation of the mucosa-adherent microbiota composition and activities within the first days of colitis resolution. Indeed, the higher relative abundance of bacteria belonging to *Lachnospiraceae*, *Eubacteriaceae* and *Bifidobacteriaceae* families, together with higher concentrations of SCFAs, the major end products of microbiota metabolic activity, may partly explain the beneficial impact of P30 diet on colon mucosa. Interestingly, the P30 diet also longitudinally increased the proportion of *Faecalibacterium*, a commensal butyrate producer, detected in healthy subjects, whose abundance is reduced in IBD patients and is positively correlated with the maintenance of clinical remission [25,26]. Butyrate is well known to be a major fuel for colonic epithelial cells, to exert pluripotent effects on the colon such as the regulation of cell growth and differentiation [27], and to present anti-inflammatory properties [28]. Butyrate deficiency in its concentrations and/or in its transport and metabolism has been indeed observed in the inflamed mucosa [29].

Although minor differences in mucosa-adherent bacterial composition were noticed between the animals fed with P14 or P30 diets in the week following the acute colitis episode (with the notable exception regarding the *Bacteroides* proportion), bacterial populations were severely modulated in P53 mice compared to the two other diets. Indeed, P53 diet consumption in DSS-treated mice decreased the proportions of butyrate-producing bacteria from *Clostridium XIVa*, *Faecalibacterium* and *Roseburia* genera in the mucosa-adherent microbiota. P53 diet-fed mice were also characterized by an increased representation of families and genera previously reported in IBD patients during flares [30,31] such as *Alloprevotella*, *Ethanoligenens*, *Klebsiella*, *Porphyromonas* and *Haemophilus*. In parallel, *Desulfovibrio*, an H_2_S producer, was reduced in P53 diet-fed mice as recently reported in the faecal microbiota recovered from UC patients [31], whereas the quantification of sulphato-reducing bacteria was not different between diets. This latter result is somewhat unexpected since an increased protein content in the diet is associated with a higher faecal H_2_S concentration [32]. However, substrate availability, microbiote composition and metabolic activity are major parameters for fixing the production of bacterial metabolites [13], and other AA-derived bacterial metabolites that are known to exert deleterious effects on intestinal epithelial cells when present in excess, such as ammonium, p-cresol or hydrogen sulphide [9,33,34,35], that might have been over-produced in P53-fed mice. Our study is in accordance with Llewellyn et al. study that shows that the level of dietary proteins has an impact on colitis severity that is associated with changes in microbiota composition [36]. In this study, the authors have shown that a high-protein diet (41%) enhanced disease progression when compared to a low-protein diet (6%). However, comparison between this study and ours should take into account the important experimental designs differences, which were the quantity and quality of proteins in the diets. Indeed, we used a mix of milk proteins containing both whey and caseins (20:80) to mimic cow milk composition, while Llewellyn et al. used casein as the only protein source. Furthermore, diets were provided one week prior to the DSS administration, whereas we started to give our different diets after colitis induction at *day 7*. Other studies, outside the scope of the present work, are necessary for determining more precisely the changes in the luminal environment in each experimental conditions. Such changes may partly explain that P53-fed mice showed a higher inflammation state but a level of epithelial crypt repair similar to P30-fed mice. Consequently, in addition to the quantity, the source of dietary proteins may modulate the IBD course [37,38] although association between protein intake and IBD risk has not been found in all studies [39,40], highlighting the urgent need for human studies to decipher the role of quantity and source of dietary proteins in IBD.

Furthermore, the reduction of carbohydrate supply in the diet may also influence colon healing. Indeed, P30 diet proportions might be more suitable to cover the requirements of macronutrients (both carbohydrates and proteins) to sustain the metabolic activity of cells [41] and synthesis of macromolecules involved in epithelial repair and restoration of barrier integrity. Additional experiments, such as the measurement of colon mucosal cell bioenergetics and protein synthesis during acute inflammation and following mucosal healing, would help to decipher these points.

## 5. Conclusions

Our results fit with the view that an optimized epithelium repair after an acute inflammatory episode can be obtained through the supply of an appropriate amount of dietary protein. It indeed depends upon a balance between the AA originating from dietary and endogenous proteins, which are available from the blood supply for mucosal healing, and the gut microbial activity towards undigested protein that enters the colon. Our study showed that a moderately high dietary protein intake during a post-inflammatory phase appears beneficial to speed up the restoration of the intestinal mucosa integrity when compared to a normoproteic diet, by affecting several key parameters associated with inflammation and healing during and after the inflammatory flare in the DSS model of colitis. In contrast, a further increase in the dietary protein amount appears counter-productive for such a process. As such, our experimental study is in accordance with the need for an increased protein intake during active IBD [7] but suggests that a threshold protein intake value should be not exceeded to avoid deleterious effects on the patients’ inflamed mucosa. 

## Figures and Tables

**Figure 1 nutrients-11-00514-f001:**
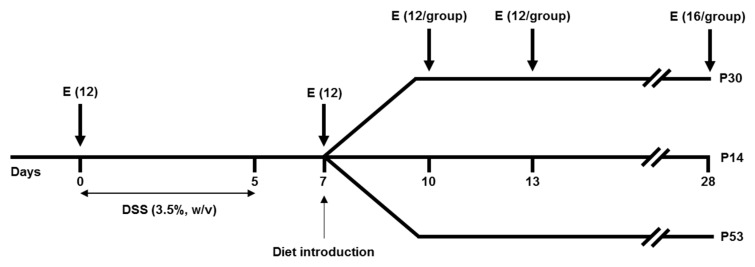
Schematic representation of the experimental design. Mice received either P14 (14% of proteins), P30 (30% of proteins) or P53 (53% of proteins) diet from D7 to D28. Mice were euthanized (E) at *day 0*, *7*, and during the resolution phase at *days 10*, *13* and *28*.

**Figure 2 nutrients-11-00514-f002:**
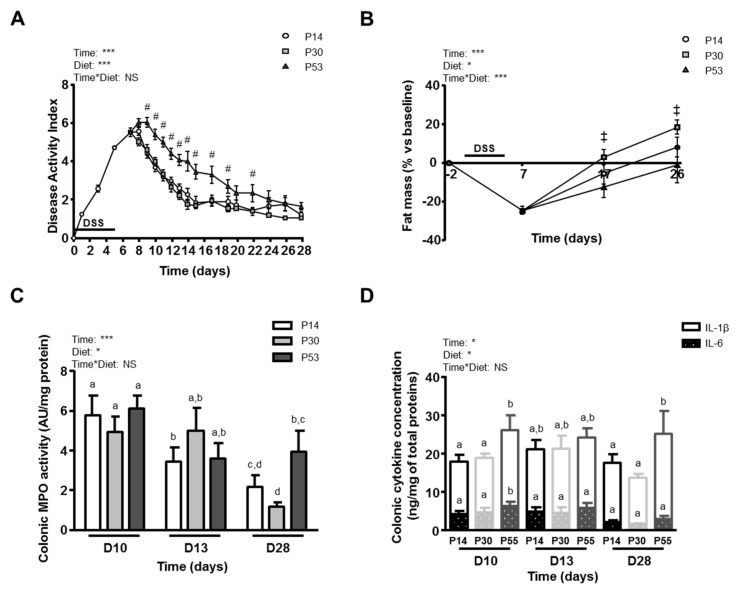
Effects of dietary protein intake level on inflammatory parameters at after 3 (D10), 6 (D13) or 21 days (D28) of dietary intervention with P14, P30 or P53 diet. (**A**): Disease activity index scoring evolution during time. #: *p* < 0.05 DSS P53 vs P14 and P30 at the same time point. (**B**): Evolution of fat mass (percentage vs baseline). ‡: *p* < 0.05 DSS P53 vs P30 at the same time point. (**C**): Colonic MPO activity. (**D**): Colonic concentrations of the pro-inflammatory cytokines IL-6 and IL-1β. Values are means ± SEM (*n* = 9–12). Means that are significantly different (*p* < 0.05) according to the post-hoc test have different letters (a or b or c or d). *: *p* < 0.05; ***: *p* < 0.001; NS: non-significant difference.

**Figure 3 nutrients-11-00514-f003:**
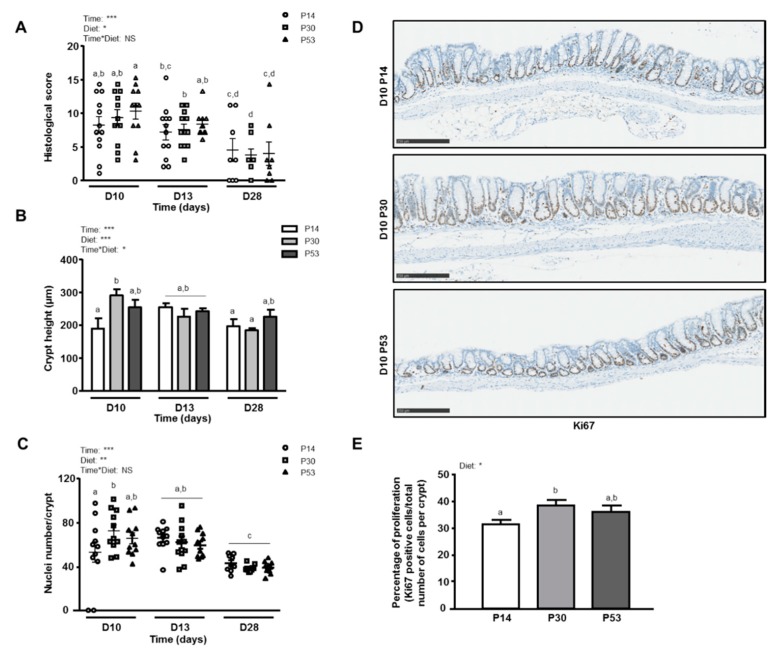
Effects of dietary protein intake on colon histomorphology after 3 (D10), 6 (D13) or 21 days (D28) of dietary intervention with P14, P30 or P53 diet. (**A**): Histological score. (**B**): Colon crypt length. (**C**): Cell numeration. (**D**). Representative illustrations of longitudinal colonic sections immunostained with an anti-Ki67 antibody. (**E**): Percentage of proliferation (Ki67-positive cells/total number of cells per crypt) at *day 10*. Values are means ± SEM (*n* = 9–12). Means that are significantly different (*p* < 0.05) according to the post-hoc test have different letters (a or b or c or d). *: *p* < 0.05; **: *p* < 0.01; ***: *p* < 0.001; NS: non-significant difference.

**Figure 4 nutrients-11-00514-f004:**
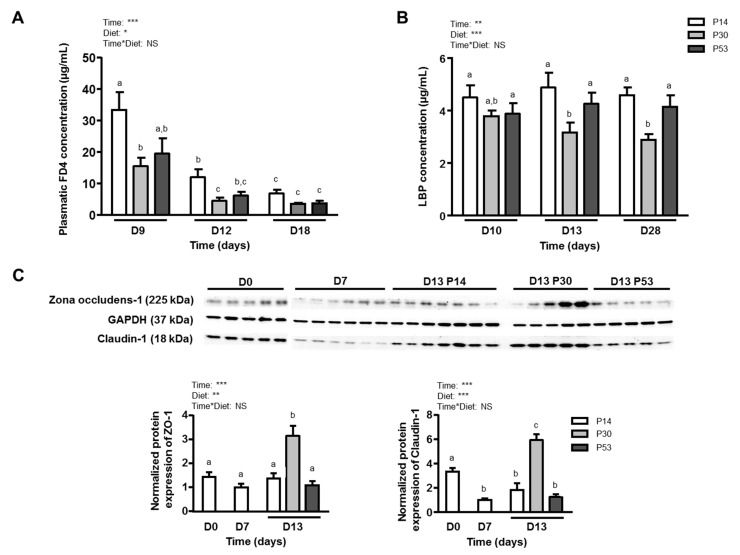
Effects of dietary protein intake level on barrier function after dietary intervention with P14, P30 or P53 diet. (**A**): Evolution of plasmatic FD4 concentration. (**B**): Plasmatic concentration of LPS-binding protein (LBP). (**C**): Western-blot analysis of the tight junction proteins Zona Occludens-1 and Claudin-1, was performed in colonic homogenates of *day 13* mice. GAPDH was used as loading control. Images are representative of four Western blots. Means that are significantly different (*p* < 0.05) according to the *post-hoc* test have different letters (a or b or c). Values are means ± SEM (*n* = 10–12). *: *p* < 0.05; **: *p* < 0.01; ***: *p* < 0.001; NS: non-significant difference.

**Figure 5 nutrients-11-00514-f005:**
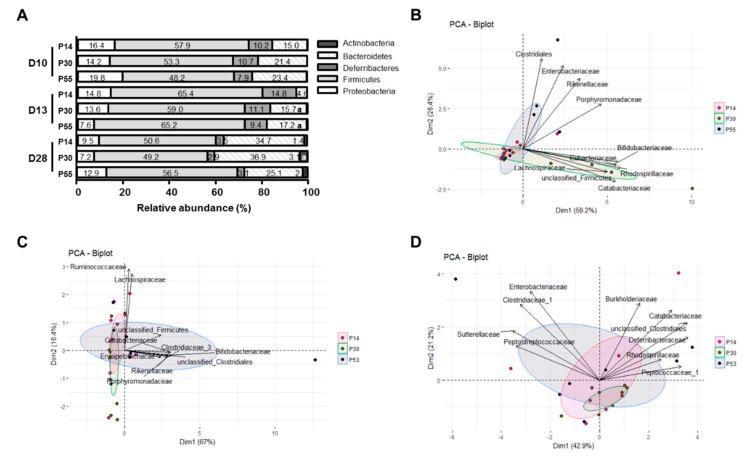
Effects of dietary protein intake on mucosa-associated microbiota composition after 3 (D10), 6 (D13) or 21 days (D28) of dietary intervention with P14, P30 or P53 diet. (**A**). Relative abundance percentages of bacterial phyla in mucosa-adherent microbiota a: vs P14, *p* < 0.05 at the same time. Due to low relative proportions, Tenericutes is not visible in the figure. (**B**–**D**): Principal coordinate analysis on family relative abundances of mucosa-adherent microbiota at *day 10* (**B**), *day 13* (**C**) and *day 28* (**D**).

**Table 1 nutrients-11-00514-t001:** Composition of the experimental diets.

	P14	P30	P53
**Weight content (g/kg)**			
Acid casein (Armor Protéines^®^, ref. #139860)	112	232	424
Whey protein (Armor Protéines^®^, Protarmor 80, ref. #139805)	28	58	96
Corn starch	622.4	493.2	287
Sucrose	100.3	79.5	45.7
Cellulose	50	50	50
Soybean oil	40	40	40
Mineral Mixture (AIN 93-M)	35	35	35
Vitamin Mixture (AIN 93-V)	10	10	10
Choline	2.3	2.3	2.3
**Energy content (%)**			
Protein	14	30	55
Carbohydrate	76	60	35
Fat	10	10	10
Energy density (kJ/g)	14.5	14.5	14.5

**Table 2 nutrients-11-00514-t002:** Effects of dietary protein intake level on colonic gene expression. Relative expression level of mRNA expressed as was means ± SEM normalized to *day 7* group (*n* = 10–12). *: *p* < 0.05; **: *p* < 0.01; ***: *p* < 0.001; NS: non-significant difference. a: vs P14, *p* < 0.05 at the same time, b: vs P53, *p* < 0.05 at the same time.

	Day 10			Day 13			Day 28			Statistical Effect		
	P14	P30	P53	P14	P30	P53	P14	P30	P53	Time	Diet	Interaction
**Epithelial Repair**												
*Saa*	1.05 ± 0.16	0.92 ± 0.21	1.33 ± 0.28	3.40 ± 0.40	4.61 ± 0.55 **a**	4.01 ± 0.68	3.18 ± 0.30	2.97 ± 0.36	2.45 ± 0.38	***	***	NS
*Il-13*	0.82 ± 0.08	1.15 ± 0.20	0.98 ± 0.16	0.78 ± 0.10	0.97 ± 0.13	0.91 ± 0.11	2.53 ± 0.39	4.42 ± 0.77 **a**	4.31 ± 0.84 **a**	***	*	NS
*Il-15*	1.67 ± 0.18	2.23 ± 0.42	3.70 ± 0.83	3.61 ± 0.91	6.78 ± 2.06 **a**	4.59 ± 1.28	4.73 ± 0.86	3.89 ± 0.67	2.37 ± 0.19	**	NS	*
*Il-22*	0.36 ± 0.08	0.61 ± 0.24	0.38 ± 0.08	0.47 ± 0.14	0.51 ± 0.15	0.57 ± 0.11	0.18 ± 0.04	0.20 ± 0.04	0.19 ± 0.04	***	NS	NS
*Il-33*	0.60 ± 0.11	0.75 ± 0.10	0.44 ± 0.11	0.43 ± 0.05	0.37 ± 0.07	0.30 ± 0.05	0.26 ± 0.06	0.18 ± 0.04	0.27 ± 0.07	***	NS	NS
*Tgf-1ß*	0.72 ± 0.07	1.01 ± 0.10 **ab**	0.60 ± 0.07	0.51 ± 0.07	0.64 ± 0.07 **b**	0.46 ± 0.05	0.43 ± 0.04	0.44 ± 0.05	0.47 ± 0.07	***	*	NS
*Tgf-ß3*	0.72 ± 0.10	0.72 ± 0.11	0.67 ± 0.09	0.93 ± 0.12	0.73 ± 0.06	0.85 ± 0.09	1.16 ± 0.16	1.04 ± 0.12	1.08 ± 0.17	**	NS	NS
*Tff3*	1.05 ± 0.12	1.63 ± 0.36	1.14 ± 0.17	1.62 ± 0.20	2.34 ± 0.35 **a**	1.70 ± 0.11	1.87 ± 0.13	2.40 ± 0.44	1.63 ± 0.23	*	**	NS
**Mucin genes**												
*Muc1*	0.58 ± 0.09	0.67 ± 0.21	0.70 ± 0.11	1.03 ± 0.14	1.41 ± 0.10	1.34 ± 0.10	0.90 ± 0.11	1.34 ± 0.17	0.75 ± 0.13	***	NS	NS
*Muc2*	1.72 ± 0.22	2.03 ± 0.21	2.37 ± 0.25	3.03 ± 0.18	3.80 ± 0.44	4.34 ± 0.37 **a**	3.38 ± 0.21	4.72 ± 0.43 **a**	5.27 ± 0.50 **a**	***	***	NS
*Muc3*	1.25 ± 0.11	1.53 ± 0.14	1.70 ± 0.15	1.73 ± 0.10	2.62 ± 0.44 **a**	2.65 ± 0.27 **a**	2.07 ± 0.27	1.94 ± 0.11	1.61 ± 0.08	***	*	*
*Muc4*	1.66 ± 0.15	2.06 ± 0.33	1.79 ± 0.18	2.32 ± 0.17	2.73 ± 0.34	3.10 ± 0.28	3.43 ± 0.41	3.36 ± 0.36	3.02 ± 0.32	**	NS	NS
**Tight-Junction Proteins**												
*Cldn1*	0.64 ± 0.10	0.84 ± 0.12 **b**	0.53 ± 0.06	0.39 ± 0.04	0.85 ± 0.17 **ab**	0.50 ± 0.11	0.38 ± 0.02	0.49 ± 0.07	0.45 ± 0.04	***	*	NS
*Cldn2*	2.77 ± 0.15	2.69 ± 0.26	3.14 ± 0.25	2.45 ± 0.33	2.34 ± 0.17	2.71 ± 0.23	2.42 ± 0.26	2.13 ± 0.21	2.69 ± 0.13	***	NS	NS
*Tjp1*	1.17 ± 0.10	1.34 ± 0.08	1.10 ± 0.07	1.30 ± 0.06	1.30 ± 0.05	1.28 ± 0.07	1.29 ± 0.06	1.09 ± 0.08	1.18 ± 0.11	**	NS	NS
*Ocln*	1.36 ± 0.11	1.52 ± 0.15	1.78 ± 0.44	1.96 ± 0.18	2.61 ± 0.25 **b**	1.89 ± 0.32	2.21 ± 0.30	2.70 ± 0.32 **b**	1.54 ± 0.12	***	*	NS

**Table 3 nutrients-11-00514-t003:** Effects of dietary protein intake level on characteristics of mucosal-adherent microbiota. Values are means ± SEM (*n* = 10–12). *: *p* < 0.05; **: *p* < 0.01; ***: *p* < 0.001; NS: non-significant difference. a: vs P14, *p* < 0.05 at the same time, b: vs P53, *p* < 0.05 at the same time.

	Day 10			Day 13			Day 28			Statistical Effect		
	P14	P30	P53	P14	P30	P53	P14	P30	P53	Time	Diet	Interaction
**Diversity and richness features**												
Shannon Index	4.56 ± 0.28	5.28 ± 0.33	4.53 ± 0.36	4.44 ± 0.26	4.23 ± 0.40	4.96 ± 0.29	5.02 ± 0.29	5.21 ± 0.19	5.31 ± 0.26	*	NS	NS
Simpson Index	0.89 ± 0.02	0.94 ± 0.01 **ab**	0.88 ± 0.03	0.88 ± 0.03	0.85 ± 0.04	0.92 ± 0.01	0.92 ± 0.02	0.93 ± 0.01	0.94 ± 0.01	***	*	NS
Chao1 Index	593 ± 31.1	678 ± 26.0	673 ± 41.1	635 ± 34.4	566 ± 25.5	642 ± 57.7	606 ± 51.7	648 ± 55.0	605 ± 39.7	**	NS	NS
Observed OTUs	286 ± 36.9	350 ± 30.1	306 ± 32.7	293 ± 24.6	335 ± 29.1 **b**	261 ± 34.8	285 ± 21.6	342 ± 30.6	305 ± 22.5	***	*	NS
**Relative genera composition (%)**												
*Alloprevotella*	0.008 ± 0.006	0.000 ± 0.000 **b**	0.079 ± 0.014 **a**	0.013 ± 0.006	0.000 ± 0.000	0.025 ± 0.025	0.003 ± 0.003	0.002 ± 0.002	0.000 ± 0.000	*	*	NS
*Bacteroides*	11.55 ± 2.771	7.391 ± 1.788 **a**	15.83 ± 4.937 **b**	3.082 ± 1.079	14.04 ± 4.541 **a**	11.92 ± 4.152	20.11 ± 4.979	21.92 ± 4.357 **b**	7.166 ± 1.920 **a**	*	*	**
*Clostridium_XIVa*	5.555 ± 2.733	2.861 ± 0.517	2.950 ± 0.763	8.545 ± 2.801	10.61 ± 3.758 **b**	3.653 ± 1.709	4.008 ± 1.635	4.212 ± 1.211	5.191 ± 1.197	*	*	NS
*Desulfovibrio*	3.062 ± 0.643	4.376 ± 0.870 **b**	1.402 ± 0.272 **a**	3.643 ± 0.946	3.892 ± 1.142 **b**	1.565 ± 0.406	1.298 ± 0.306	1.699 ± 0.581	0.939 ± 0.207	**	**	NS
*Enterobacter*	0.021 ± 0.010	0.013 ± 0.008	0.025 ± 0.019	0.049 ± 0.020	0.023 ± 0.019	0.025 ± 0.013	0.031 ± 0.015	0.012 ± 0.005	0.015 ± 0.008	NS	NS	NS
*Enterorhabdus*	0.010 ± 0.005	0.009 ± 0.004	0.006 ± 0.003	0.004 ± 0.003	0.028 ± 0.016	0.034 ± 0.012 **a**	0.030 ± 0.014	0.023 ± 0.008	0.014 ± 0.008	*	*	NS
*Ethanoligenens*	0.000 ± 0.000	0.000 ± 0.000	0.000 ± 0.000	0.000 ± 0.000	0.000 ± 0.000	0.000 ± 0.000	0.000 ± 0.000	0.000 ± 0.000 **b**	0.013 ± 0.013 **a**	NS	*	NS
*Faecalibacterium*	0.000 ± 0.000	0.002 ± 0.002	0.000 ± 0.000	0.004 ± 0.004	0.007 ± 0.002 **b**	0.001 ± 0.001	0.014 ± 0.014	0.048 ± 0.008 **b**	0.000 ± 0.000	**	*	NS
*Haemophilus*	0.000 ± 0.000	0.004 ± 0.004 **b**	0.017 ± 0.003 **a**	0.000 ± 0.000	0.013 ± 0.007	0.025 ± 0.012 **a**	0.002 ± 0.002	0.004 ± 0.004 **b**	0.043 ± 0.029 **a**	NS	*	NS
*Klebsiella*	0.004 ± 0.005	0.000 ± 0.000 **b**	0.005 ± 0.001	0.000 ± 0.000	0.000 ± 0.000	0.000 ± 0.000	0.002 ± 0.002	0.000 ± 0.000 **b**	0.005 ± 0.001	*	*	NS
*Porphyromonas*	0.000 ± 0.000	0.000 ± 0.000	0.000 ± 0.000	0.000 ± 0.000	0.000 ± 0.000	0.000 ± 0.000	0.000 ± 0.000	0.000 ± 0.000 **b**	0.012 ± 0.012 **a**	**	**	NS
*Roseburia*	0.005 ± 0.005	0.009 ± 0.004	0.010 ± 0.005	0.011 ± 0.006	0.006 ± 0.003	0.011 ± 0.008	0.009 ± 0.006	0.029±0.004 **ab**	0.011 ± 0.001	***	***	NS
**Caecal SCFA** (µmol/g)												
Total	0.06 ± 0.04	2.05 ± 1.56 **a**	0.38 ± 0.27	0.14 ± 0.07	1.32 ± 0.44 **a**	0.99 ± 0.46	4.88 ± 1.08	6.85 ± 1.65 **b**	12.32 ± 3.65 **a**	***	*	NS
Acetate	0.04 ± 0.04	1.40 ± 1.04 **a**	0.28 ± 0.18	0.14 ± 0.07	1.20 ± 0.43 **a**	1.00 ± 0.38	3.40 ± 0.71	4.66 ± 1.13	8.38 ± 2.42	***	*	NS
Propionate	0.00 ± 0.00	0.44 ± 0.36 **a**	0.07 ± 0.07	0.00 ± 0.00	0.00 ± 0.00	0.13 ± 0.13	0.90 ± 0.26	1.31 ± 0.35	2.55 ± 0.65	***	*	NS
Butyrate	0.01 ± 0.01	0.21 ± 0.16 **a**	0.03 ± 0.03	0.00 ± 0.00	0.12 ± 0.05	0.10 ± 0.07	0.59 ± 0.13	0.88 ± 0.26	1.40 ± 0.39	***	*	NS

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
