# Peer review of "Dietary Protein Intake Level Modulates Mucosal Healing and Mucosa-Adherent Microbiota in Mouse Model of Colitis"

_nutrients, 2019, doi:10.3390/nu11030514_

Reviewer 1 Report

Article "Dietary protein intake level modulates mucosal healing and mucosa-adherent microbiota in mouse model of colitis" explained well and how dietary protein alleviates through microbiota by improving colonocytes tight junctions, cytokine levels, microbiota levels etc. Authors well executed the study and provided adequate data to support their findings. However, some of the details or description needed to support to their work further which as follows below;

1. It is highly helpful if the authors shows any serum and/or fecal biomarkers levels like Lipocalin-2, etc., in order to validate the p30 diet mediated improvement of gut microbiota to reduce inflammation.

2. Authors are requested to give justification regarding the controversial observation by recently published work and validate how different from their observation. 

Llewellyn, S. R., Britton, G. J., Contijoch, E. J., Vennaro, O. H., Mortha, A., Colombel, J. F., ... & Faith, J. J. (2018). Interactions between diet and the intestinal microbiota alter intestinal permeability and colitis severity in mice. Gastroenterology154(4), 1037-1046.

Like the above article reported that the concentration they used for protein diet which is similar concentration of the current paper (p30), showed enhanced the disease progression and make the microbiota improve susceptibility to colitis. They also observed the serum cytokine levels also changed opposite to the current observation.

Author Response

We would like to thank the referee for their helpful comments. You will find below point by point responses to all comments raised and related modifications in the revised version of our manuscript.

Reviewer 1:

Article "Dietary protein intake level modulates mucosal healing and mucosa-adherent microbiota in mouse model of colitis" explained well and how dietary protein alleviates through microbiota by improving colonocytes tight junctions, cytokine levels, microbiota levels etc. Authors well executed the study and provided adequate data to support their findings.

We thank reviewer 1 for positive comment

However, some of the details or description needed to support to their work further which as follows below;

Point 1: It is highly helpful if the authors shows any serum and/or fecal biomarkers levels like Lipocalin-2, etc., in order to validate the p30 diet mediated improvement of gut microbiota to reduce inflammation.

Response 1: We agree with this referee’s comment about the interest to measure plasmatic or fecal biomarkers of inflammation. We actually have also studied systemic effects of inflammation and protein diet intake level by measuring plasmatic SAA and IL-6 concentrations as well as body mass composition. These data will be included in another manuscript dedicated to the effects of the dietary protein content on protein metabolism in different tissues. The results presented in this other paper show a positive effect of P30 diet on systemic inflammation. Concerning measurement of fecal biomarkers, we indeed initially planned to quantify fecal lipocalin-2. Unfortunately, we could not collect enough fecal material in DSS-treated mice since they suffered from diarrhea particularly after day 7 and we used all colon luminal contents for determination of water percentage and osmolarity.

Point 2: Authors are requested to give justification regarding the controversial observation by recently published work and validate how different from their observation.

Llewellyn, S. R., Britton, G. J., Contijoch, E. J., Vennaro, O. H., Mortha, A., Colombel, J. F., ... & Faith, J. J. (2018). Interactions between diet and the intestinal microbiota alter intestinal permeability and colitis severity in mice. Gastroenterology, 154(4), 1037-1046.

Like the above article reported that the concentration they used for protein diet which is similar concentration of the current paper (p30), showed enhanced the disease progression and make the microbiota improve susceptibility to colitis. They also observed the serum cytokine levels also changed opposite to the current observation.

Response 2: Thank you for this interesting comment. The differences between the study of Llewellyn et al. and our present study might be explained by the experimental design and the composition of the diets that were not identical, since in Llewellyn’s study, they mostly compared high casein diet (44% of proteins) to low casein diet (6%, not used in our present study). The protein composition was also slightly different since we used a mix of milk proteins containing both whey and caseins (20:80) to mimic cow milk composition unlike study of Llewellyn, who used casein as the sole source of protein (while whey isolate in this study did not show any effect on DSS-colitis aggravation, although protein level was low). Diets were provided one week prior to the administration of DSS whereas we started to give our different diets after colitis induction at day 7. As Llewellyn’s study, we also previously showed that protein intake level impacted DSS-course with a high protein diet (53%) being more deleterious for mice (with higher body weight loss, mortality and inflammatory markers) when given simultaneously with DSS, than a 14% protein diet (Lan et al, 2016 ), but we did not test in this previous study other protein levels (such as 30 % (P30), equivalent to MC CEL diet of Llewellyn’ study, although the results using this level of protein was little commented by the authors in this study). However, our study that use a different experimental design is in accordance with the Llewellyn’ study major findings that “dietary casein concentration on disease severity is microbiota-dependent”. Indeed, we found in our study major differences with different levels of protein on both microbiota and colitis. We have modified our discussion to include this point in the revised version of our manuscript.

We again thank the referee for giving us the opportunity to modify and ameliorate our manuscript.

We hope that the revised version of our manuscript will be suitable for publication in Nutrients.

Pending your response, we send you our best regards.

Annaïg LAN (Corresponding author)

Reviewer 2 Report

This is a meticulous study with a large amount of clearly presented data. The clinical significance of the results to human IBD (approximately  - moderate protein good, high or low bad) is arguably overemphasised. Specific issues:

Introduction - it is incorrect  to say "strong consensus recommending an increase i protein intake in active IBD" and this is not what ref 7 says.

There could be a more critical appraisal of the literature on protein intake and IBD - there is reasonably strong evidence incriminating a harmful effect of high meat intake in ulcerative colitis (but probably not in Crohn's disease) - so the source of protein intake might be more important than the total amount.

It is always very difficult to extrapolate from DSS colitis to human IBD - arguably it is more like UC than CD but response to therapy in mouse DSS colitis has often not been translated into man. Moreover it is of course not really feasible to study human omnivore diets in a mouse model.

Both points 2 and 3 could be considered in discussion and more caution used in trying to extrapolate to human disease.

I note that ANOVA and Bonferroni corrections seem to have been used appropriately but there are a huge number of endpoints and many of the dietary differences look fairly modest . Arguably there should be more focus on the hypothesis/primary endpoint which seems to have been the impact on barrier function (see last para in Introduction) - the data here eg fig 4 do look convincing.

Author Response

We would like to thank the referee for his helpful comments. Please, find below point by point responses to all comments raised and related modifications in the revised version of our manuscript.

Reviewer 2:

This is a meticulous study with a large amount of clearly presented data.

We thank the reviewer for this positive comment.

Point 1: The clinical significance of the results to human IBD (approximately - moderate protein good, high or low bad) is arguably overemphasised. Specific issues:

Introduction - it is incorrect to say "strong consensus recommending an increase protein intake in active IBD" and this is not what ref 7 says.

Response 1: We agree with the reviewer that this sentence needs to be somewhat moderated. This point has been corrected in the revised version.

Point 2 and 3: There could be a more critical appraisal of the literature on protein intake and IBD - there is reasonably strong evidence incriminating a harmful effect of high meat intake in ulcerative colitis (but probably not in Crohn's disease) - so the source of protein intake might be more important than the total amount. It is always very difficult to extrapolate from DSS colitis to human IBD - arguably it is more like UC than CD but response to therapy in mouse DSS colitis has often not been translated into man. Moreover it is of course not really feasible to study human omnivore diets in a mouse model. Both points 2 and 3 could be considered in discussion and more caution used in trying to extrapolate to human disease.

Response 2 and 3: We agree with these referee’s comments about these two points and in particular the difficulty of extrapolation of colitis murine model to human IBD. These points have been added in the revised version.

Point 4: I note that ANOVA and Bonferroni corrections seem to have been used appropriately but there are a huge number of endpoints and many of the dietary differences look fairly modest. Arguably there should be more focus on the hypothesis/primary endpoint which seems to have been the impact on barrier function (see last para in Introduction) - the data here eg fig 4 do look convincing.

Response 4: We agree with the referee that some of the differences induced by diets observed in our study look modest. Probably because of the strong effect of DSS-induced inflammation on barrier function compared to dietary interventions which have often a slight impact.

We again thank the referee for giving us the opportunity to modify and ameliorate our manuscript.

We hope that the revised version of our manuscript will be suitable for publication in Nutrients.

Pending your response, we send you our best regards. 

Annaïg LAN (Corresponding author)